# Microwave-Assisted Synthesis, Biological Activity Evaluation, Molecular Docking, and ADMET Studies of Some Novel Pyrrolo [2,3-*b*] Pyrrole Derivatives

**DOI:** 10.3390/molecules27072061

**Published:** 2022-03-23

**Authors:** Moumen S. Kamel, Amany Belal, Moustafa O. Aboelez, E. Kh. Shokr, H. Abdel-Ghany, Hany S. Mansour, Ahmed M. Shawky, Mahmoud Abd El Aleem Ali Ali El-Remaily

**Affiliations:** 1Chemistry Department, Faculty of Science, Sohag University, Sohag 82524, Egypt; hossameldeen.hussien@science.sohag.edu.eg; 2Department of Pharmaceutical Chemistry, College of Pharmacy, Taif University, P.O. Box 11099, Taif 21944, Saudi Arabia; 3Department of Pharmaceutical Chemistry, Faculty of Pharmacy, Sohag University, Sohag 82524, Egypt; 4Physics Department, Faculty of Science, Sohag University, Sohag 82524, Egypt; eshokr@yahoo.com; 5Department of Medicinal Chemistry, Faculty of Pharmacy, Assiut University, Assiut 71526, Egypt; hany_samy2011@yahoo.com; 6Science and Technology Unit (STU), Umm Al-Qura University, Makkah 21955, Saudi Arabia; amesmail@uqu.edu.sa

**Keywords:** diphenyl-1, 6-dihydropyrrolo [2,3-*b*]pyrrole, hypocholesterolemic, hypotriglyceridemic activities, microwave irradiation, TC, TGs, LDL and ABTS method

## Abstract

Novel pyrrolo [2,3-*b*] pyrrole derivatives were synthesized and their hypolipidemic activity was assessed in hyperlipidemic rats. The chemical structures of the new derivatives were confirmed through spectral analysis. Compounds **5** and **6** were revealed to be the most effective hypolipidemic agents, with considerable hypocholesterolemic and hypotriglyceridemic effects. They appear to be promising candidates for creating new powerful derivatives with anti-atherosclerotic and hypolipidemic properties. As for antimicrobial activity, some of the tested compounds showed moderate activity against *Pseudomonas aeruginosa*: compound **2** revealed an MIC value of 50 μg/mL, compared to 25 μg/mL for ciprofloxacin. Compound **3** showed good antimicrobial activity against Staphylococcus aureus, comparable to ciprofloxacin, and roughly half the activity of ampicillin, according to MIC values. Compound **2** has an MIC approximately 25% of that of clotrimazole against Candida albicans. Compound **2** also showed the highest antioxidant activity with 59% inhibition of radical scavenging activity. Additionally, the cytotoxic activity of these new derivatives **1**–**7** was investigated and most of them showed good anticancer activity against the three tested cell lines.

## 1. Introduction

Pyrroles are five-membered heterocyclic organic compounds; their annular structure consists of four carbon atoms and one nitrogen atom with the formula of C_4_H_5_N. Several pyrrole derivatives constructed from natural sources or synthesized in the laboratory have different biological activities. For example, many pyrrole derivatives have shown biological activities such as antimicrobial [1], antimycobacterial [2], and antioxidant activity [3]. They may also be used as anticancer [4], anti-inflammatory [5], and hypolipidemic activities [6]. Several studies on pyrrole derivatives have reported on their insecticidal and acaricide effectiveness against various pests of agricultural importance and public health [7,8,9].

Hyperlipidemia is a major risk factor for atherosclerosis and its complications, including coronary heart disease (CHD), ischemic cerebrovascular disease, and peripheral vascular disease [10]. Cardiovascular diseases (CVDs) are considered to be the cause of more than 30% of all deaths globally. By 2020, they were expected to be the top cause of death worldwide. A 10% reduction in serum cholesterol has been shown to reduce the incidence of CHD by 30% [11,12,13].

The Niemann–Pick C1-like 1 protein, on the other hand, is another target for lowering plasma lipids (NPC1L1). NPC1L1 [14,15,16] is a transmembrane protein present in enterocytes and hepatocytes. It carries sterol for either intestinal or hepatobiliary cholesterol absorption and excretion.

The 3D structure of NPC1L1 reveals a cholesterol binding site with multiple extracellular and intracellular loops at its N-terminal domain, whereas ezetimibe binds to the C-terminal extracellular domain and disrupts the configurational changes required for NPC1L1 activity and cholesterol binding [16].

The suggested compounds were produced and tested for their capacity to improve lipid profile, as well as in silico research to see if they may potentially bind to the NPC1L1 active site, which is projected to be the leading cause of mortality globally.

In the present study, synthesized novel fused pyrrole derivatives will be investigated for their diverse biological activities as antihyperlipidemic, antimicrobial, antifungal and antioxidant activity.

## 2. Result and Discussion

### 2.1. Chemistry

The reaction of 2-(bis(ethylthio)methylene)malononitrile (i) with aniline in a 1:2 ratio produced 2-(bis(phenylamino)methylene)malononitrile (ii). The IR spectra revealed a new absorption band corresponding to the NH group at 3430 and 3400 cm^−1^ (Figure 1). The presence of exchangeable NH as a singlet at 9.45 ppm and multiple signals at 7.42–7.28 for phenyl groups is confirmed by its ^1^H NMR spectra. Diethyl 3,4-diamino-1,6-diphenyl-1,6-dihydropyrrolo[2,3-*b*]pyrrole-2,5-dicarboxylate **1** [17,18] was reacted with formamide, formic acid, isatin, phenylisocyanate, 2,5-dimethoxytetrahydrofurane and metylamine to yield the corresponding compounds **2**–**7** (Figure 2).

IR spectra of compounds **2**–**7** showed the disappearence of *N*H_2_ group absorption bands and showed new C=O_amide_ NH groups absorption bands at 1678 and 3327, 3442 cm^−1^, respectively, while the ^1^H-NMR spectrum of compound **4** (Appendix A) showed new signals corresponding to exchangable NH and CH_aromatic_ at 11.55, 7.42 and 6.57 ppm. The ^13^C-NMR spectrum showed new signals for C=O groups at δ 184.84, 163.72, 162.05, and 159.83 ppm. The mechanism formation of compound **7** was postulated to proceed via nucleophile attack of amino group at =CH– with two ethanol molecules that were eliminated to form Schiff bases. Then, there was another nucleophilic attack of methylamine at the N=CH− group to form imino, which was followed by nucleophilic attack of the remaining NH group at CO ester with elimination of ethanol molecule (Figure 3).

### 2.2. Biological Activities

#### 2.2.1. Hypolipidemic Activity

The hypolipidemic activity of the synthesized compounds was investigated in a high cholesterol diet (HCD)-fed hyperlipidemic rat model versus a hyperlipidemic control (rats fed with HCD and given drug vehicle % carboxymethylcellulose (CMC) only by oral administration of 20 mg/kg of the tested compounds) HCD (enriched with 2% cholesterol and 1% cholic acid) was used to produce hyperlipidemia in rats [19]. Regular control rats were fed normal rodent chow and the pharmaceutical carrier (0.5% CMC) [20,21]. The results were compared to those achieved with gemfibrozil, a commonly used antihyperlipidemic drug (Table 1).

As rats were fed a high-cholesterol diet for 14 days, the serum levels of TC, TGs, and LDL increased by 56.50%, 68.56%, and 101.50%, respectively, when compared to normal control rats. Furthermore, inducing hyperlipidemia reduces blood HDL levels by 33.55% compared to normal control rats. Cholic acid, which enhances fat and cholesterol absorption from the intestine, was added to a high-cholesterol diet to improve the sensitivity of rats to hyperlipidemia [21].

The compounds **5**, **6**, and **7** considerably lowered the serum TC level of hyperlipidemic rats by 19.45%, 14.7%, and 9.17%, respectively, and were more active than the standard gemfibrozil. Furthermore, the compound **4** had antihyperlipidemic action (5.46%) that was comparable to the reference. Similarly, there was no significant reduction in serum TC levels in the groups treated with compounds **2**, **3**, and **4** (Table 1, Figure 1).

The compounds **6** and **7** lowered serum TG levels in hyperlipidemic rats by 28.77% and 24.45%, respectively, and are more active than the standard gemfibrozil, which only reduced serum TG levels by 22.68%. Furthermore, compounds **2**, **4**, and **7** elicited rather minor action (Table 1, Figure 2).

Compounds 5 and 6 significantly increased HDL levels in hyperlipidemic rats’ serum, ranging from 33.47% to 41.13% higher than the control, while compounds **2**, **4**, and **7** have activity similar to that of the standard (20.43%, 17.30% and 21.11%, respectively Table 1, Figure 3). Compounds **5** and **6** were found to be more active than gemfibrozil in lowering the serum level of LDL in hyperlipidemic rats by percentages ranging from 16 to 16.85% (Table 1, Figure 4).

The lowest LDL/HDL ratio was found in compounds **5** and **6** (4.13, 4.31). This important finding highlights the function of the substances described in modifying lipid profiles and avoiding atherosclerosis and associated cardiovascular problems [22].

Finally, the analysis of different groups’ lipid profile data (TC, TG, HDL, and LDL) clearly indicates that some of the produced compounds have good hypolipidemic activity. Compounds **5** and **6** appear to be effective hypolipidemic drugs that work through a variety of pathways. They resulted in significant reductions in serum levels of TC, TG, and LDL, as well as an increase in serum HDL and reduced LDL/HDL ratios. The top chemicals in this group are compounds **5** and **6**.

#### 2.2.2. Screening of Antibacterial and Antifungal Activities

Compounds **1**–**7** were tested for antimicrobial activity in vitro against four microorganisms: E. Gram-positive bacteria including *E. coli* ATCC8739 and Pseudomonas aeruginosa ATCC 9027 (−Ve) bacteria, Staphylococcus aureus ATCC 6583P as Gram (±Ve) bacteria, and Candida albicans ATCC 2091 as a yeast-like fungus using the diffusion method [23]. Compounds **1**–**7** were made at concentrations of 200, 100, 50, 25, and 12.5 μg/mL, respectively. The lowest concentration required to inhibit bacterial growth (MIC) was computed and listed in the inhibition zones (Table 2).

#### 2.2.3. In Vitro Cytotoxic Activities

The MTT test was used to assess the antiproliferative activity of the target compounds **1**−**7** against a panel of three cancer cell lines: MCF-7, HCT-116 and A549. In this in vitro screening, erlotinib and doxorubicin were used as controls. The results are shown as IC_50_ (µM) in Table 3. Compounds **1**–**7** were shown to be more effective than erlotinib against the tested cell lines. In MCF-7, HCT-116, and A549 cells, compound 2 revealed IC_50_ values of 3.81, 2.90, and 2.39 times more active than erlotinib, respectively. Compound 2 was also more effective on MCF-7 and HCT-116 cells than doxorubicin, and compound 2 proved to be the most effective one against MCF-7 cells.

#### 2.2.4. Antioxidant Activity by ABTS Method

The 2,2-azinobis(3-ethyl-benzothiazoline-6-sulfonic acid)diammonium salt (ABTS) technique was used to test the antioxidant activity of compounds **1**–**7** [24]. The ABTS assay is based on antioxidants’ ability to scavenge the long-lived ABTS± radical cation. Sodium persulfate is used to convert ABTS to its radical cation. The hue of this radical cation is blue, and it absorbs light at 734 nm. Most antioxidants, including phenolics, thiols, and vitamin C, react with the ABTS radical cation. The blue ABTS radical cation is transformed to its colorless neutral form during this reaction. Spectrophotometric monitoring of the reaction is possible. The Trolox equivalent antioxidant capacity (TEAC) assay is another name for this test.

Figure 5 illustrates the percent inhibition of radical scavenging activity for investigated compounds **1**–**7**. Compound **2** showed highest antioxidant activities with 59% inhibition of radical scavenging activity, while compounds **3**–**5** have moderate antioxidant activity (48–51%). Compounds **3** and **7** have mild antioxidant activity (36–38%).

## 3. Molecular Docking Study

Molecular modeling, especially docking, has made significant contributions to the search for possibly novel active molecules as a new technique for rationalizing, predicting the behavior of molecules at molecular level, and investigating the most appropriate conformations and interactions of hit ligand(s) at the binding site of the macromolecular protein of interest. The main goal is to forecast how our produced chemicals will affect cholesterol transport protein (Niemann–Pick C1-like 1 protein) inhibition. The revelation of structural information for the Niemann–Pick C1-like 1 protein structure (NPC1L1, pdb code: 3QNT)3 and docking of Ezetimibe, a therapeutically marketed inhibitor of this protein 4 were the catalysts for this. Figure 6A,B show the binding mechanisms of Ezetimibe with the active site of this transport protein.

Docking of compounds **1**–**7** into the active site of the Niemann–Pick C1-like 1 protein (NPC1L1, pdb code: 3QNT) was carried out using the commercial program Molecular Operating Environment (MOE) 2019.01 in order to examine the fitting modes and binding interactions at the 3QNT active site. The obtained results of docking, including docking scores and examination of the 2D and 3D poses, manifested that the designed ligands are well fitting into the active site and interact through favorable binding forces with the essential amino acid residues at the targeted protein’s binding site, revealing that they could be considered virtually as cholesterol absorption inhibitors. The compounds almost formed stable complexes with the enzyme active site (energy score 8.73 to 8.26; Table 3), supporting our hypothesis that inhibiting cholesterol absorption by the NPC1L1 enzyme is responsible for some of the antihyperlipidemic effects reported. All of the substances were able to bind to the ezetimibe binding site in the enzyme6’s N-terminal domain. Figure 6 and Table 4 show how Ezetimibe interacts with the NPC1L1 active site by hydrogen bonding between the carbonyl of Glu38 and the phenolic OH in Ezetimibe and another hydrogen bonding between the NH of imidazole ring of His124 and carbonyl O of lactam ring in Ezetimibe, as well as by hydrophobic interaction (H-pi interaction) between the aromatic imidazole ring of His124.

The general binding patterns of the designed derivatives showed interaction with His124, which is reported to be a crucial residue in NPC1L1 binding site. Though all molecules form stable complexes at the binding site, that does not clarify the variable activity observed. The number of interactions found and overall binding score of each molecule can offer such explanation. In terms of the binding score, compounds **2**, **5** and **6** have the lowest binding scores, which are lower than Ezetimibe itself; this reveal predicted good activity as inhibitors of NPC1L1. The investigated ligands displayed similar binding forces compared to those exhibited with Ezetimibe, which strengthens the hypothesis that both have a similar mode of action as NPC1L1 inhibitors and consequently inhibitors of cholesterol absorption. Compound **2** bonded with three significant hydrogen bonds, Glu38, Gln95 and His124, with bond lengths less than 3 Ǻ and with one hydrophobic interaction with Pro215 of the active site (binding score = −8.48 Kcal/mol), Figure 7 and Table 4. Compound **5** also showed a high number of interactions with the NPC1L1 active site with a binding score = −8.51 Kcal/mol. The interactions between compound **5** and the active site are represented by two hydrogen bonds with Ser51 and Ser98 amino acid residues with bond lengths of 2.84 and 3.23 Ǻ, respectively, and two hydrophobic interactions with Gln95 and His124 with bond lengths of 4.92 and 3.81 Ǻ, respectively, Figure 8 and Table 4. Finally, compound 6 interacted with the active site by three interactions and two hydrogen bonds with Ser98 and Asn127 where the drug acts as the H-acceptor and the amino acid residues act as H- bond donors. The third interaction in compound **6** is a hydrophobic pi–pi interaction between the N-phenyl moiety of the ligand and the imidazole ring of the amino acid residue, Figure 9 and Table 4. The previously mentioned interactions indicate the importance of aromatic moieties, the H-acceptor and H-donor in the designed ligands; this also maintains an appropriate lipophilicity of the designed compound to introduce appropriate pharmacokinetics. In summary, the designed molecules can be considered as promising candidates for the treatment of hyperlipidemia through the inhibition of NPC1L1 and consequently inhibition of cholesterol absorption, though further testing is still required, including in vitro testing and assessment of pharmacokinetic parameters in in vivo models**.**

## 4. In Silico Pharmacokinetics and Toxicological Profile

The ADME/Tox profile is used to describe the absorption (A), distribution (D), metabolism (M), excretion (E), and toxicity (Tox) of drugs. This in silico pharmacokinetics profile is a beneficial tool to predict the ADME/Tox properties of investigated drug candidates, especially in pre-clinical levels. To make efficient pharmacokinetics and toxicological predictions, online available and in silico models have been set up. The established models are helpful for drug improvement and avoiding late-stage drug candidate termination; therefore, they conserve the considerable non-productive utilization of time and money [25]. Two relatively recent web tools were used in this in silico study: the costless accessible SwissADME web tool (http://www.swissadme.ch/) (accessed on 25 January 2022) [26], which is a relevant and recent computational software to predict the ADME/Tox properties of small compounds [27], and the freely available pkCSM-pharmacokinetics web tool (http://biosig.unimelb.edu.au/pkcsm/prediction) (accessed on 27 January 2022) [28], which is a new technique for evaluating and optimizing small-molecules’ ADME/Tox profile and depends on experimental data and graph-based signatures [29]. The structures of the investigated molecules (**1**–**7**) and Ezetimibe were expressed in SMILES (simplified molecular-input line-entry specification) to be used in the utilized web tools, SwissADME and pkCSM-pharmacokinetics. The predicted absorption was evaluated from gastrointestinal absorption, aqueous solubility, lipophilicity, and percentage of human intestinal absorption properties. All the investigated molecules possess estimated high GI absorption and are thus predicted to have good intestinal absorption (like the control molecule, Ezetimibe and have an intermediate to good score for oral bioavailability. Additionally, drug bioavailability was linked to the topological polar surface area (TPSA); if the TPSA of certain molecule is equal to or less than 140 Å^2^, this molecule will have an estimated good bioavailability in the experimental animal [30]. The range of TPSA values for the investigated molecules is from 72.32 to 145.38 Å^2^, supporting the probability of having predicted high passive oral absorption. The majority of the studied molecules exhibited water solubility ranging between −4.5 and −8.18, which implies moderate to poor water solubility compared to Ezetimibe which has water solubility = −4.92. Lipophilicity values were estimated through the logarithm of the n-octanol/water partition coefficient, which was evaluated utilizing the consensus LogP_o/w_ parameter of SwissADME. LogP_o/w_ is strongly linked to transport approaches, involving human membrane permeability and distribution to the variable organs and tissues [31]. Ordinarily, for a certain molecule to have good human oral bioavailability, it should possess a moderate logP (0 < logP < 3) [31]. For our evaluated molecules, the estimated values of logP_o/w_ ranged from 2.63 to 6.76, Table 5. To support the SwissADME estimation, pkCSM-pharmacokinetics was utilized to estimate the percentage of intestinal absorption in humans. pkCSM-pharmacokinetics reveals that all the studied molecules possess an absorption percentage exceeding 91%, which is similar to that of Ezetimibe (91.8%), Table 5.

Distribution was estimated utilizing the glycoprotein P (P-gp) substrate and blood–brain barrier (BBB) permeation by SwissADME and fraction unbound parameters by pkCSM-pharmacokinetics. All molecules had no BBB permeation, while compounds **1**, **3**, **5** and **6** are not substrates for the P-gp efflux pump and compounds **2**, **4**, **7**, and Ezetimibe are not subjected to the P-gp efflux pump mechanism; consequently, they are unable to cross the BBB, and thus they are predicted to have no CNS side effects. Normally, there is an equilibrium between the bound and unbound state to serum proteins such as albumin. The unbound fraction of molecules to serum proteins in the plasma affects renal glomerular excretion and hepatic biotransformation, and thus affects the volume of distribution, total drug clearance, and effectiveness of drugs [32].

The higher the plasma protein binding of the drug in the blood, the less effectively it can distribute among/cross cell membranes [29]. In this work, the unbound fraction values were estimated for the eight studied molecules involving Ezetimibe in human plasma to have low values, ranged from 0.063 to 0.387.

The metabolism of the investigated molecules was predicted utilizing the SwissADME web tool through studying the possibility of these molecules to inhibit the main cytochrome (CYP 450) enzymes, CYP1A2, CYP2C19, CYP2C9, CYP2D6 and CYP3A4. CYP enzymes are among the main enzymes involved in xenobiotic metabolism in the body specifically through oxidation; thus, the inhibition of these enzymes may lead to metabolism-related drug–drug interactions, usually including competition for the same enzyme binding site with another co-administered drug. Enzyme inhibition inhibits the metabolism and elimination of other co-administered xenobiotics, leading to greater plasma levels of these co-administered drugs that affect the therapeutic value of these drugs. Thus, this inhibition of CYPs can deteriorate and lead to xenobiotic toxicity or at least loss of the therapeutic benefits of a drug [33]. In addition to Ezetimibe, all investigated molecules were evaluated to not inhibit CYP1A2. All molecules and Ezetimibe probably inhibit CYP2C19 except molecules **2**, **4**, **5** and **7**. In contrast to the other compounds, compounds **2**, **5** and Ezetimibe were predicted to not inhibit CYP2C9. In contrast to Ezetimibe, all compounds are likely to inhibit CYP2D6. CYP3A4 appeared not to be inhibited by compounds **1**, **2**, **5** and 7. Significantly, compounds **2**, **5** and 7 have the least effect as inhibitors of the tested cytochrome enzymes.

Excretion as a pharmacokinetic property is a conjunction of renal and hepatic excretion, is linked to bioavailability, and is critical for determining doses in rates to obtain steady-state concentrations for certain drugs [29]. Excretion estimation was achieved through calculating the total clearance and Renal OCT2 substrate parameters using pkCSM-pharmacokinetics web tool. Firstly, the total clearance values ranged from −0.041 to 1.198 (log mL/min/kg), which is far away from total clearance value of Ezetimibe (−0.269). Secondly, all the investigated compounds and Ezetimibe are not predicted substrates for Renal OCT2 transport protein, Table 5. Lastly, the prediction of the hepatotoxicity and oral rat acute toxicities of the investigated molecules was reached through pkCSM-pharmacokinetics [29]. Calculated LD_50_ values are in between 2.208 to 3.727 mol/kg and are greater than that of Ezetimibe (1.908 mol/kg). The liver is the main organ for the metabolism of drugs and xenobiotics, and it also has a critical role in energy exchanges. Consequently, a liver experiencing damage will affect normal biotransformation and could even result in liver failure and cirrhosis [34]. The hepatotoxicity parameter estimated that all the studied compounds and Ezetimibe may experience hepatotoxicity.

## 5. In Silico Target Prediction of Compounds 1–7 and Ezetimibe

In recent years, good support to predict the most possible targets for small molecules can be reached by constructed bio-/chemo-informatics software techniques. This target speculation, which is called ligand-based target prediction, has displayed high performance and the possibility for quick prediction of macromolecular targets for certain small molecules in drug-discovery strategies [35]. The prediction of different macromolecular targets was accomplished by utilizing the SwissTarget Prediction web tool (http://www.swisstargetprediction.ch/, accessed on 10 February 2022) [36] in trials aiming at the prediction of the most probable macromolecular targets of small compounds. This target prediction analysis was limited to the top 15 homo sapien macromolecular targets of kinase, enzymes and family A G protein-coupled receptors as drug targets appeared in the top 15 targets with variable percentages of all tested compounds and Ezetimibe. Compounds 5 and 6 have a high percentage of kinases and enzymes as drug targets, Figure 10. The percentages of kinase, enzymes and family A G protein-coupled receptors as drug targets for all the tested compounds are illustrated in Figure 11.

## 6. Experimental Section

### 6.1. Chemistry

#### 6.1.1. Materials and Methods

Merck (München, Germany) provided all of the reagents, which were used without additional purification. Thin-layer chromatography (TLC) was used to monitor all reactions, with precoated plates of silica gel G/UV-254 of 0.25 mm thickness (Merck 60F254) and UV light (254 nm/365 nm) for visibility. Uncorrected melting points were detected using Kofler melting points equipment. The attenuated total reflection (ATR) method was used to record infrared spectra with an FT-IR-ALPHBROKER-Platinum-ATR spectrometer. In DMSO-d_6_, ^1H^-NMR and ^13^C-NMR spectra for all compounds were obtained at 400 MHz and 100 MHz, respectively, on a Bruker Bio Spin AG spectrometer. Chemical shifts (HZ) were measured in parts per million (ppm) using tetramethylsilane (TMS) as an internal standard (=0) for ^1^H NMR. Chemical shift, integration, and multiplicity (s = singlet, d = doublet, t = triplet, q = quartet, m = multiplet) were expressed as coupling constants (J) in hertz (Hz). On a Perkin–Elmer CHN-analyzer model, elemental analyses were obtained. The microwave irradiation was carried out on a SINEO Microwave Chemistry Technology Co., Ltd., Shanghai, China. MAS-II Plus microwave synthesis/extraction reaction workstation from their Microwave Chemistry reaction platform.

#### 6.1.2. Synthesis of Diethyl 3,4-Diamino-1,6-diphenyl-1,6-dihydropyrrolo[2,3-b]pyrrole-2,5-dicarboxylate (1)

We added 0.3 mol, 41.5 g oven-dried potassium carbonate into 40 mL DMF and 0.1 mol, 6.6 g malononitrile dissolved in 20 mL DMF to a combination of 0.3 mol, 41.5 g oven-dried potassium carbonate in 40 mL DMF and 0.1 mol, 6.6 g malononitrile dissolved in 20 mL DMF (0.15 mol, 9.0 mL). Drop by drop, under vigorous stirring, carbon disulfide was added. The mixture was cooled to 0 °C after 30 min, and 0.2 mol, 15.5 ml of ethyl iodide in 10 mL of DMF was added for 20 min. The reaction mixture was heated for 3 min at 900 W in a microwave before being emptied into 200 mL of cold water and washed three times with 100 mL of water. Ethyl bromoacetate (0.2 mol, 16.7 ml) and DMF (20 ml) were heated in a microwave for 5 min at 900 W, and then poured into cold water. The precipitate was recovered, washed three times with water, dried, and ethanol crystallized. Yield 62%, yellow powder, m.p. 140 °C; FT-IR (KBr, Cm^−1^): 3438, 3342 (2NH_2_), 3077 (aromatic), 2978 (aliphatic), 1724 (2CO_ester_) cm^−^^1^, H-NMR (DMSO-*d_6_*): δ 7.48–7.46 (t, *J =* 7.4 Hz, 6H_aromatic_), 7.30–7.29 (d, *J =* 7.3 Hz, 4H_aromatic_), 5.99 (s,4H, 2NH_2_), 3.94–3.89 (q, *J =* 3.91 Hz, 4H, 2CH_2ester_), 0.88–0.84 (t, *J =* 3.91 Hz, 6H, 2CH_3ester_); ^13^CNMR: δ 162.1, 153.1, 140.9, 130.8, 129.5, 124.4, 121.7, 111.7, 58.5, 14.6 ppm; for chemical formula: C_24_H_24_N_4_O_4_; elemental analysis: C, 66.65; H, 5.59; N, 12.96. Found: C, 66.62; H, 5.61; N, 12.94.

#### 6.1.3. Synthesis of Compounds (**2** and **3**)

General procedure: Compound **1** (0.432 g, 1 mmol) and excess formamide and formic acid (10 ml) were warmed for 10 min in a microwave at 900 W. After allowing the mixture to cool, it was treated with ice-cold water. Filtered solid product was obtained, which was then washed with water, dried, and crystallized from ethanol.

#### 6.1.4. 5,6-Diphenyl-3,5,6,8-tetrahydropyrimido[4′′,3′′:4′,5′]pyrrolo[3′,2′:4,5]pyrrolo[3,2-d]pyrimidine-4,7-dione (**2**)

Yield 81%, white powder, mp > 350 °C; FT-IR (KBr, Cm^−1^): 3464, 3330 (2NH), 3035 (aromatic), 1655 (2CO); ^1^H-NMR (DMSO-*d_6_*): δ 12.79 (s, 2H, 2NH), 8.33 (s, 2H, 2CH_Het._), 7.75–7.53 (m, 10H_aromatic_); ^13^C-NMR: δ 152.12, 148.84, 145.56, 144.64, 136.42, 129.48, 129.04, 128.72, 119.42, 114.04, 110.94; for chemical formula: C_22_H_14_N_6_O_2_ (394); elemental analysis: C, 67.00; H, 3.58; N, 21.31. Found: C, 67.03; H, 3.56; N, 21.30.

#### 6.1.5. Diethyl 3,4-Diformamido-1,6-diphenyl-1,6-dihydropyrrolo[2,3-b]pyrrole-2,5-dicarboxylate (**3**)

Yield 77%, white powder, mp > 350 °C; FT-IR (KBr, Cm^−1^): 3452, 3336 (2NH), 2930 (aliphatic), 1712 (2CO_ester_), 1655 (2CO); ^1^H-NMR (DMSO-*d_6_*): δ 11.55 (s, 2H, 2NH), 7.88 (s, 2H, 2CHO), 7.53–7.31 (m, 10H_aromatic_), 4.06–4.01 (q, *J =* 3.8 Hz, 4H, 2CH_2ester_), 1.03–1.00 (t, *J =* 3.8 Hz, 6H, 2CH_3ester_); ^13^C-NMR: δ 162.28, 159.25, 158.56, 150.05, 141.87, 137.77, 129.79, 129.14, 128.65, 120.77, 113.57, 113.24; for chemical formula: C_26_H_24_N_4_O_6_ (488); elemental analysis: C, 63.93; H, 4.95; N, 11.47. Found: C, 63.91; H, 4.94; N, 11.45.

#### 6.1.6. Synthesis of Diethyl 3,4-bis(2-Oxoindolin-3-ylidene)amino)-1,6-diphenyl-1,6-dihydropyrrolo[2,3-b]pyrrole-2,5-dicarboxylate (**4**)

A mixture of compound **1** (0.432 g, 1 mmol) and isatin (0.249 g, 2 mmol) was heated under reflux for 10 min. The crystalline product was collected, washed with cold ethanol and crystallized from ethanol.

Yield 82%, green powder, mp > 320 °C; FT-IR (KBr, Cm^−1^): 3448, 3387 (2NH), 3085 (aromatic), 2973 (aliphatic), 1730 (2CO_ester_), 1665 (2CO); ^1^H-NMR (DMSO-*d_6_*): δ 10.97 (s, 2H, 2NH), 7.61–7.48 (m, 12H_aromatic_), 7.30–7.29 (d, *J =* 7.6 Hz, 2H_aromatic_), 7.09–7.06 (d, *J =* 8.3 Hz, 2H_aromatic_), 6.93–6.91 (t, *J = * 7.6 Hz, 2H_aromatic_), 3.92–3.89 (q, *J = * 3.9 Hz, 4H, 2CH_2ester_), 0.88–0.84 (t, *J =* 3.9 Hz, 6H, 2CH_3ester_); ^13^C-NMR: δ 184.8, 160.0, 159.8, 151.2, 146.5, 138.9, 129.7, 128.8, 125.1, 123.3, 119.8, 118.2, 114.7, 112.7, 108.8, 59.5, 14.0; for chemical formula: C_40_H_30_N_6_O_6_ (690); elemental analysis: C, 69.56; H, 4.38; N, 12.17. Found: C, 69.56; H, 4.35; N, 12.16.

#### 6.1.7. Synthesis of 3,5,6,8-Tetraphenyl-1,5,6,10-tetrahydropyrimido [4′′,3′′:4′,5′]pyrrolo[3′,2′:4,5]pyrrolo[3,2-d]pyrimidine-2,4,7,9(3H,8H)-tetraone (**5**)

In pyridine (15 mL), a mixture of compound 1 (0.432 g, 1 mmol) and phenyl isocyanate (0.239 g, 2 mmol) was heated under reflux for 12 h. After cooling, the mixture was placed into a mixture of ice-cold water and HCl solution. Filtered solid product was obtained, which was then washed with water, dried, and crystallized from ethanol.

Yield 76%, yellow powder, mp = 340 °C; FT-IR (KBr, Cm^−1^): 3324, 3278 (2NH), 3045 (aromatic), 1717, 1694 (4CO); ^1^H-NMR (DMSO-*d_6_*): δ 8.64 (s, 2H, 2NH), 7.50–7.48 (t, *J* = 8.5 Hz, 12H_aromatic_), 7.29–7.27 (d, *J* = 8.7 Hz, 6H_aromatic_), 6.98–6.96 (d, *J* = 8.7 Hz, 2H_aromatic_); ^13^C-NMR: δ160.0, 158.9, 153.0, 146.5, 140.4, 129.4, 129.2, 128.9, 128.7, 122.2, 121.9, 121.6, 121.0, 118.6; for chemical formula: C_34_H_22_N_6_O_4_ (578); elemental analysis: C, 70.58; H, 3.83; N, 14.53. Found: C, 70.54; H, 3.86; N, 14.50.

#### 6.1.8. Synthesis of Diethyl 1,6-diphenyl-3,4-di(1H-pyrrol-1-yl)-1,6-dihydro pyrrolo[2,3-b]pyrrole-2,5-dicarboxylate (**6**)

In 10 mL of glacial acetic acid, a mixture of compound **1** (0.432 g, 1 mmol) and 2,5-dimethoxytetrahydrofuran (0.264 ml, 2 mmol) was heated under reflux for 4 h. The liquid was allowed to cool before being emptied into ice cold water and filtered, dried, and crystallized from ethanol. Yield 73%, brown powder, mp > 320 °C; FT-IR (KBr, Cm^−1^): 3058 (aromatic), 2973 (aliphatic), 1711 (2CO_ester_); ^1^H-NMR (DMSO-*d_6_*): δ 8.42 (t, *J =* 13.5 Hz, 6H, CH_aromatic_), 8.22–8.21 (d, *J =* 7.5 Hz,2H_aromatic_), 7.33–7.31 (d, *J =* 13.3 Hz,4H, 4CH_Het_), 6.13–6.12 (t, *J =* 7.6 Hz, 4H _Het_), 4.28–4.23 (q, *J =* 3.9 Hz, 4H, 2CH_2ester_), 1.29–1.24 (t, *J =* 3.9 Hz, 6H, 2CH_3ester_); ^13^C-NMR: δ 160.28, 158.85, 151.05, 142.87, 129.79, 129.14, 128.65, 120.77, 114.57, 114.24, 91.34, 61.43, 14.07; anal. calc. for chemical formula: C_32_H_28_N_4_O_4_ (532); elemental analysis: C, 72.17; H, 5.30; N, 10.52. Found: C, 72.14; H, 5.29; N, 10.50.

#### 6.1.9. General Procedure of Synthesis 3,8-Dimethyl-5,6-diphenyl-3,5,6,8-tetrahydro-pyrimido [4′′,3′′:4′,5′]pyrrolo[3′,2′:4,5]pyrrolo[3,2-d]pyrimidine-4,7-dione (**7**)

The reaction mixture was concentrated under vacuum after heating a mixture of compound **1** (0.432 g, 1 mmol) and triethyl orthoformate (5 ml) under reflux for 3 h. A total of 2 mL methyl amine was added and the mixture was heated at 40 °C for 3 h. The reaction mixture was diluted with aqueous ethanol (1:1), and the precipitated crystals were filtered, dried, and recrystallized from ethanol. Yield 74%, mp > 350 °C; FT-IR (KBr, Cm-1): 3439 (NH), 3057 (aromatic), 2984 (aliphatic), 1756 (CO); 1HNMR (DMSO-d6): δ 8.37 (s, 2H aromatic), 7.55–7.53 (t, J = 6.9 Hz, 6H aromatic), 7.47 -7.45 (d, J = 7.0 Hz, 4H aromatic), 3.41 (s, 3H, CH3), 2.47 (s, 3H, CH3); 13CNMR: δ 158.12, 148.84, 145.56, 136.42, 129.84, 129.04, 128.72, 119.42, 114.04, 92.07, 33.73, 18.16; anal. calc. for chemical formula: C24H18N6O2 (422); elemental analysis: C, 68.24; H, 4.29; N, 19.89. Found: C, 68.23; H, 4.27; N, 19.87.

### 6.2. Biological Activities

#### 6.2.1. Hypolipidemic Activity

A total of 42 male adult Wistar albino rats weighing 170 mg were used in this study. Animals were purchased from the Faculty of Science’s animal home in Sohag, Egypt, and were fed commercial rat chow and tap water as a regular diet. Rats were given a week to acclimate to their new surroundings before being used in the experiment. Rats were kept under typical laboratory settings, with 12 h light/12 h dark cycles and an ambient temperature of 25 °C. The animals had free access to food and water for up to 24 h before being used.

Hyperlipidemia was generated in rats by giving them a high-cholesterol diet (HCD) made by mixing normal rodent chow with 4% cholesterol and 1% cholic acid (*w*/*w*) (Sigma, CA, USA) for 30 days in a row [19].

The rats were placed into seven groups, each with six rodents. The normal control group was fed a conventional meal and given a drug vehicle of 0.5% carboxymethylcellulose (CMC) orally. HCD was given to the remaining six groups for 30 days. Group II was treated with gemfibrozil orally at a dose of 20 mg/kg per day as a conventional hypolipidemic treatment (CMC) of 0.5% as a drug vehicle; group III was treated with gemfibrozil orally at a dose of 20 mg/kg per day as a hyperlipidemic control [22]. Compounds **1**, **2**, **4**, and **5** were given orally to rats in groups IV, V, VI, and VII at a dose of 20 mg/kg/day (1/10 of the LD_50_ dose), calculated earlier using the Lorke method [21]. The treatment in the last six groups began 15 days following the onset of hyperlipidemia induction and lasted 15 days [19].

Rats from all groups were starved for 16 h before being slaughtered by cervical dislocation and blood samples were taken at the end of the experiment. The serum was separated by centrifugation at 3000 rpm for 10 min at 4 °C and stored at −20 °C until analysis. Commercially accessible diagnostic kits (Randox, Crumlin, UK) were used to estimate total cholesterol (TC), triglycerides (TGs), and high-density lipoprotein cholesterol (HDL-C) levels using a Jenway UV-vis spectrophotometer (Jasco spectropho- tometer USA). The method was carried out according to the manufacturer’s instructions, and the TC, TGs, and HDLc concentrations were measured in mg/dl. Friedewald’s formula [37] was used to calculate the concentration of low-density lipoprotein cholesterol (LDL-C) in the blood.

#### 6.2.2. Antimicrobial and Antifungal Activity

Compounds **1**–**7** were evaluated using the diffusion method against the Gram (−Ve bacteria Escherichia coli ATCC8739, Pseudomonas aeruginosa ATCC 9027, Gram (±Ve) bacteria Staphylococcus aureus ATCC 6583P, and yeast-like fungus Candida albicans ATCC 2091 [23]. Bacteria were grown on Nutrient Agar (NA) medium, while fungi were grown on Sabouraud Dextrose Agar (SDA). Pouring NA or SDA (60 mL) into Petri plates (150 mm 15 mm) and letting it harden was used to prepare them. After drying the plates, 1 mL of each standardized inoculum suspension was poured and evenly distributed.

The surplus inoculums were drained, and the inoculums were dried for 15 min. A sterile cork borer (6 mm in diameter) was used to make eight equidistant wells in the medium, and 75 L of the test chemicals (1 mg/mL) diluted in *N,N*-dimethylformamide (DMF) was poured into the wells. The tests were performed three times. Standard antibacterial and antifungal medicines were ampicillin trihydrate (10 μg/disc), ciprofloxacin (5 μg/disc), and clotrimazole (100 μg/disc). As a control, it was incubated. The plates were incubated for 24 h at 37 degrees Celsius. The average diameter of bacterial growth inhibition zones around the discs in mm was reported for each tested chemical.

MIC measurements [23] were performed using the twofold serial dilution approach for drugs that demonstrated significant inhibitory zones. Compounds **1**–**7** were made at concentrations of 200, 100, 50, 25, and 12.5 μg/mL, respectively. Antibacterial and antifungal activity was determined using the microdilution susceptibility test in Muller–Hinton broth (oxoid) and Sabouraud Liquid Medium (oxoid), respectively. The produced test compounds were inoculated in the previously described serial dilution broth with microbe suspensions at 106 CFU/mL (colony forming unit/mL). The culture tubes were incubated for 24 h at 37 degrees Celsius. The development of bacteria was observed by turbidity measurements at the end of the incubation period [38]. The MIC is the lowest concentration at which no bacterial growth was observed.

#### 6.2.3. In Vitro Cytotoxic Activity Screening

The antiproliferative characteristics of the target compounds 1–7 were tested in vitro against MCF-7, HCT-116 and A549 cell lines [38,39,40] using the MTT assay technique. The ATCC (American Type Culture Collection) cell lines were obtained from the Holding business for biological products and vaccines (VACSERA) in Cairo, Egypt. The following is how the antiproliferative activity was measured. Human cancer cell lines were seeded in 96-well plates at a density of 3–8 × 10^3^ cells per well. In a 5 percent CO_2_ incubator, the wells were then incubated for 12 h at 37 °C. The culture medium was replaced with 0.1 mL of new medium containing graded amounts of the test chemicals for each well to determine the DMSO level. The wells have a two-day incubation period. The cells were subsequently grown for another 4 h in each well in 100 L MTT solutions (5 g mL^−1^). After MTT-formazan crystals were dissolved in 100 L DMSO, the absorbance of each well was measured at 490 nm using an automated ELISA reader system (TECAN, CHE). Nonlinear regression fitting models were used to find the IC_50_ values Graph Pad, Prism Version 5. The results were based on the average of three separate, duplicate trials and were expressed as means SD.

#### 6.2.4. Antioxidant Activity by ABTS Method

The 2,2-azinobis(3-ethyl-benzothiazoline-6-sulfonic acid)diammonium salt (ABTS) technique was used to test the antioxidant activity of compounds **2**–**7** [24]. Chemically, an ABTS± radical cation stock solution was created by combining ammonium peroxodisulfate solution (0.2 mL, 65 mmol/dm^3^) with ABTS solution (50 mL, 1 mmol dm^3^, prepared in 0.1 mol dm^3^ phosphate buffer pH 7.4). The combination was left overnight, and then the absorbance at 734 nm (AABTS) was measured by mixing ABTS ± radical cation stock solution (0.5 mL) with phosphate buffer (2 mL, pH 7.4) in a cuvette. After that, test compounds (0.5 mL, 0.5 mmol) were added to the cuvette, the solution was immediately mixed, and the absorbance at 734 nm (compound) was measured after 60 s.

The decrease in absorbance caused by the addition of Trolox as the standard was measured using the same procedure for each concentration of Trolox (50–600 mol/L), and the calibration curve for the decrease in absorbance (A = AABTS − ATrolox) of Trolox vs. Trolox concentration was constructed using linear regression. On the basis of the Trolox calibration curve, the antioxidant activity of the chosen compounds was estimated and expressed in mol/L of Trolox equivalents (mol TE/L).

## 7. Molecular Modeling

The molecular docking studies were performed using Molecular Operating Environment MOE 2019, 01; Chemical Computing Group, Montreal QC, Canada as the computational software. Briefly, docking of the investigated molecules was performed in four steps: preparation of the 3D structure of the target and its active site, preparation of the ligands, running docking, and interpretation of the results. The X-ray three-dimensional crystal structure of Niemann–Pick C1-like 1 protein (NPC1L1) was acquired from the protein data bank, PDB (PDB code 3QNT. The target structure was prepared by structure preparation order through a running protonate 3D panel, correct order then fixing the protein structure. The active site was found and isolated. Structures of the molecules were built utilizing the MOE builder interface then energy minimization for all structures to an RMS (root mean square) distance of 0.1 Å and RMSD (root-mean-square deviation) gradient of 0.01 kcal/mol. Docking of the molecular database of the designed ligands was carried out using the MOE-DOCK software wizard. Molecular docking was completed through adjusting the default settings of the MOE program. (Placement is triangular Matcher; Rescoring 1 is London dG with retain equals 30; Refinement is Forcefield; Rescoring 2 is GBVI/WSA dG with retain equals 30) To compare the investigated ligands, the binding affinity of the docked molecules was expressed as binding score (S, kcal/mol); the lower S-values indicate a more favorable pose.

## 8. In Silico ADME/Tox Profile of the Synthesized Compounds (1–7) and Ezetimibe

The freely accessible SwissADME (http://www.swissadme.ch/) (accessed on 25 January 2022) [26] and pkCSM-pharmacokinetics (http://biosig.unimelb.edu.au/pkcsm/prediction) (accessed on 27 January 2022) web tools were used in this predictive study, which are recent methods for estimating and optimizing small-molecule pharmacokinetics and toxicity profiles [29]. The structures of the investigated molecules (**1**–**7**) and Ezetimibe were built on ChemDraw Ultra 8.0 and then copied as SMILES nomenclature, then pasted into the used web tools of interest, SwissADME or pkCSM-pharmacokinetics. The most important pharmacokinetics and toxicity parameters found in the web tools were selected to represent the pharmacokinetics and toxicity profile. The results are given in Table 4.

## 9. In Silico Target Prediction of the Synthesized Compounds (1–7) and Ezetimibe

The freely accessible Swiss Target Prediction web tool (http://www.swisstargetprediction.ch, accessed on 10 February 2022) [39] web tool was used in this predictive study and the top **15** targets option was chosen on the online program webpage. The molecular structures of the most potent compounds (**1**–**7**) and Ezetimibe were built on ChemDraw Ultra 8.0 and then copied as SMILES (simplified molecular-input line-entry specification) nomenclature, then pasted into the used web tool, Swiss Target Prediction. The results are given in Figure 11.

## 10. Conclusions

Due to diverse biological activities that were previously reported for fused and substituted pyrrole derivatives, a series of fused pyrroles and pyrimidine-substituted pyrroles and indole-substituted pyrroles were synthesized and subjected to structural confirmation through spectral analysis. Investigating the diverse biological activities for these new derivatives was explored and revealed that compounds **5** and **6** are hopeful antihyperlipidemic agents, and molecular docking studies were supportive of the hypolipidimic activity. Moderate antibacterial activity was reported for these new derivatives. Pyrroles fused with pyrimidine **2** showed antifungal activity equal to 25% of clotrimazole activity and also proved to be the most potent antioxidant agent. Additionally, the new derivatives revealed a good potential cytotoxicity profile against all the tested cell lines.

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
