# Peer review of "Microwave-Assisted Synthesis, Biological Activity Evaluation, Molecular Docking, and ADMET Studies of Some Novel Pyrrolo [2,3-b] Pyrrole Derivatives"

_molecules, 2022, doi:10.3390/molecules27072061_

Round 1
Reviewer 1 Report
The work presented to me for review is interesting, but in my opinion, it needs to be supplemented to be published in Molecules. Primarily, compounds with potential biological use should be tested for their cytotoxicity, which is missing in this work. This should be supplemented.
The methods used and the results obtained were well described by the authors, but they did not avoid several minor errors that required correction.
Examples of errors:
line 71 - missing spaces after NH2
line 72 - unnecessary space
line 68 - "Scheme 2" should be written before the period
line 80 - as above
the signature of Scheme 2 - compounds, not compound
signatures of schemes, figures, and tables should start with a capital letter
Figures 1, 2, 3, 4 - a different font would be better, such as in the text
Figures 5 and 11 are of poor quality
links throughout the manuscript appear too large (examples line 250, 253)
the elements in figure 10 are too small, it is impossible to read them
the volume unit should always be written the same, either ml or mL
References should be edited by the journal guidelines
In my opinion, the manuscript could be accepted for publication in Molecules only after all changes have been made.
Author Response
Thank you for your email received on 22 Feb 2022, we have revised the manuscript carefully and addressed all comments raised by the reviewer; Hope the manuscript after improvement is now suitable for publication in Molecules.
Here are our detailed responses to the comments:
Dear Sir; First we would like to thank you for the valuable comments and advice.
Please, enclose here with our response to the comments on the Manuscript ID: molecules-1612668 which is also included through the text of the attached revised form of Manuscript.
- line 71 - missing spaces after NH2
Response: We have modified this in manuscript.
- line 72 - unnecessary space
Response: We have modified this in manuscript.
- line 68 - "Scheme 2" should be written before the period
Response: We have modified this in manuscript.
- line 80 - as above
Response: We have modified this in manuscript.
- The signature of Scheme 2 - compounds, not compound
Response: We have modified this in manuscript.
- signatures of schemes, figures, and tables should start with a capital letter
Response: We have modified this in manuscript.
- Figures 1, 2, 3, 4 - a different font would be better, such as in the text
Response: We have modified this in manuscript.
- Figures 5 and 11 are of poor quality
Response: We have modified this in manuscript.
- links throughout the manuscript appear too large (examples line 250, 253)
Response: We have modified this in manuscript.
- The elements in figure 10 are too small, it is impossible to read them
Response: We have modified this in manuscript.
- The volume unit should always be written the same, either ml or mL
Response: We have modified this in manuscript.
- References should be edited by the journal guidelines
Response: We have modified this in manuscript.
Reviewer 2 Report
The manuscript by Kamel and co-workers reports the preparation of novel pyrrolo [2, 3-b] pyrrole derivatives and the evaluation of the antihyperlipidemic, antimicrobial, antifungal and antioxidant activities .
Overall the manuscript is of interest in the field of medicinal chemistry, but some issues should be addressed for the pubblication in "molecules" journal.
The introduction provides a poor background about reported pyrrole derivatives and too general. the authors should focus on a specific biological activity. Moreover, the insertion of a figure with the chemical structures of described pyrroles should be useful for the readers.
The chemistry section of "results and discussion" is very confusing. the compounds in the schemes are called 1,2 etc, but also i, ii ( 2 is the aniline and also a final compound). the authors should edit the text.
A minor English check is required.
Author Response
Thank you for your email received on 22 Feb 2022, we have revised the manuscript carefully and addressed all comments raised by the reviewer; Hope the manuscript after improvement is now suitable for publication in Molecules.
Here are our detailed responses to the comments:
- The introduction provides a poor background about reported pyrrole derivatives and too general. the authors should focus on a specific biological activity. Moreover, the insertion of a figure with the chemical structures of described pyrroles should be useful for the readers.
Response: Thanks for this valuable comment; we have modified this in manuscript.
- The chemistry section of "results and discussion" is very confusing. the compounds in the schemes are called 1,2 etc, but also i, ii ( 2 is the aniline and also a final compound). the authors should edit the text.
Response: We have modified this in manuscript.
Round 2
Reviewer 1 Report
Unfortunately, the authors completely ignored one of my comments regarding cytotoxicity. Only minor editorial errors have been corrected. Therefore, I still believe that the manuscript needs to be supplemented.
Author Response
To: Editorial Board, Molecules
ID: molecules-1612668
Dear Professor Cora Zhang,
Thank you for your email received on 4 March 2022, we have revised the manuscript carefully and addressed the comments raised by the reviewer; Hope the manuscript after improvement is now suitable for publication in Molecules.
Here are our detailed responses to the comments:
(Reviewer 1)
Unfortunately, the authors completely ignored one of my comments regarding cytotoxicity. Only minor editorial errors have been corrected.
Response: Dear respected reviewer, really we are so sorry for that, it is unintentional at all, it was by mistake and we appreciate your valuable comments that helped us in improving our manuscript. We have performed the required experiments and incorporated their results in the revised manuscript. They are highlighted in the attached revised manuscript.
Thanks in return.
Amany Belal